# Novel Pitolisant-Derived Sulfonyl Compounds for Alzheimer Disease

**DOI:** 10.3390/ijms25020799

**Published:** 2024-01-08

**Authors:** Silvia Pérez-Silanes, Eva Martisova, Esther Moreno, Maite Solas, Daniel Plano, Carmen Sanmartin, María Javier Ramírez

**Affiliations:** Department of Pharmaceutical Sciences, Universidad de Navarra, Irunlarrea 1, 31008 Pamplona, Spain; sperez@unav.es (S.P.-S.); emartisova@alumni.unav.es (E.M.); esthermorenoamatria@gmail.com (E.M.); msolaszu@unav.es (M.S.); dplano@unav.es (D.P.); sanmartin@unav.es (C.S.)

**Keywords:** pitolisant-derived sulfonylureas, Alzheimer disease, acetylcholine release, novel object recognition test, blood brain barrier permeability

## Abstract

Alzheimer’s disease (AD) is a complex and multifactorial neurodegenerative disorder characterized by cognitive decline, memory loss, behavioral changes, and other neurological symptoms. Considering the urgent need for new AD therapeutics, in the present study we designed, synthesized, and evaluated multitarget compounds structurally inspired by sulfonylureas and pitolisant with the aim of obtaining multitarget ligands for AD treatment. Due to the diversity of chemical scaffolds, a novel strategy has been adopted by merging into one structure moieties displaying H_3_R antagonism and acetylcholinesterase inhibition. Eight compounds, selected by their binding activity on H_3_R, showed a moderate ability to inhibit acetylcholinesterase activity in vitro, and two of the compounds (derivatives **2** and **7**) were also capable of increasing acetylcholine release in vitro. Among the tested compounds, derivative **2** was identified and selected for further in vivo studies. Compound **2** was able to reverse scopolamine-induced cognitive deficits with results comparable to those of galantamine, a drug used in clinics for treating AD. In addition to its efficacy, this compound showed moderate BBB permeation in vitro. Altogether, these results point out that the fragment-like character of compound **2** leads to an optimal starting point for a plausible medicinal chemistry approach for this novel strategy.

## 1. Introduction

Alzheimer’s disease (AD) is a neurodegenerative illness that affects people as they age. It causes behavioral changes, forgetfulness, memory loss, and impairments in language and cognition. By 2040, neurodegenerative diseases will surpass cancer-related deaths as the second-most common cause of death in developed nations, with cardiovascular disease-related deaths coming in second [1]. Of the three cases of AD, two occur in women [2]. Across nearly all age groups, there are more women than men with AD [3]. Women typically experience the following differences from men: a 70% higher lifetime cost of living with the disease [4], increased rates of brain atrophy [5,6], faster rates of cognitive decline [5,7], and later diagnosis [8]. Additionally, more than 60% of caregivers for people with AD or related dementias are women [9]. According to mounting data, women going through the menopause between the ages of 45 and 55 may be the most vulnerable to developing AD [10]. Menopause symptoms are repeatedly and consistently linked to poor outcomes in women’s brain health [11,12,13,14,15,16,17]. Unsurprisingly, women have a one in five lifetime risk of developing AD by age 45, compared to a one in ten risk for age-matched men [10].

Cholinergic neuronal degeneration, tau protein phosphorylation, amyloid cascade hypothesis, neuroinflammation, decreased glucose utilization, calcium theory, oxidative stress, mitochondrial dysfunction, altered insulin signaling, dysregulation of iron metabolism, and abnormal cholesterol homeostasis are just a few of the numerous pathogenic mechanisms that have been linked to AD. According to the cholinergic hypothesis, AD is correlated with the brain’s acetylcholinesterase (AChE) enzyme’s hydrolysis of acetylcholine (ACh). As AD disease advances, there is a rapid decrease in ACh levels and an increase in free radical levels.

Currently, symptomatic treatments for AD include acetylcholinesterase inhibitors (AChEI) such as donepezil, rivastigmine, galantamine, tacrine, and NMDA receptor blockers such as memantine [18]. AChE decomposes ACh to acetate (CH_3_COO^−^) and choline (Ch), which triggers the termination of neurotransmission in brain synapses. A lowered ACh level is believed to be the primary cause of the cognitive deficiency observed in AD. On the other hand, histamine H_3_ receptor (H_3_R) antagonist/inverse agonists can act as precognitive agents by increasing the levels of histamine and other neurotransmitters, including ACh [19]. Pitolisant is the first and only FDA- and EMA-approved H_3_R antagonist/inverse agonist [20]. Initially, Pitolisant was approved in 2016 in Europe for use in the treatment of narcolepsy in adults and later in the USA and Canada [21]. It was able to reduce sleepiness in narcoleptic patients [22]. Pitolisant has a broad range of potential therapeutic applications for central nervous system (CNS) disorders, including Alzheimer’s and neuropsychiatric diseases [20]. So, pitolisant has been suggested as a potential drug target for the treatment of these illnesses.

Moreover, studies have reported some sulfonylureas, such as glimepiride, to be an effective competitive AChEI [23]. In addition, the UW-MD-72 derivative was developed and characterized as a reversible and competitive potent AChEI and as an H_3_R antagonist with a high selectivity profile, ameliorating cognitive impairments via mechanisms dependent on cholinergic muscarinic neurotransmission [24]. More recently, the literature has described promising H_3_R compounds with cholinesterase inhibition activities with beneficial therapeutic effects in AD [25,26] (Figure 1).

Based on these investigations, we have synthesized several derivatives structurally inspired by sulfonylureas and pitolisant with the aim of obtaining multitarget ligands for AD treatment. We selected these skeletons that have been modified with different aromatic moieties in the sulfonylurea core. In this study, related to the pitolisant feature, we have processed a set of three carbons linked with different secondary cyclic amines such as morpholine, piperidine, or pyrrolidine.

## 2. Results

### 2.1. Chemistry

The novel pitolisant-derived sulfamide and sulfonylurea derivatives were prepared as previously reported [27]. Figure 1 summarizes the synthesis procedure of studied derivatives **1**–**8**.

### 2.2. Biological Evaluation

#### 2.2.1. Acetylcholinesterase Inhibition

Compounds were selected as H_3_ receptor antagonists (0.08–0.5 μM, as described in [27]). Based on their binding activity on H_3_R [27], eight compounds were selected for determining their AChE inhibitory activity (Table 1). Even though all the compounds showed inhibitory activity, only two of them, **2** and **7**, showed inhibitory activity higher than 37% at the screening concentration of 1 µM. These two compounds were selected for subsequent studies.

A detailed study on the ability of compounds **2** and **7** (0.01–10 µM) to inhibit AChE in vitro was next performed. Galantamine is one of the three AChEIs currently used as therapeutic agents for the treatment of AD, and it was included in this study as a reference drug. All three compounds were able to significantly inhibit AChE activity in a concentration-dependent manner (Figure 2). However, compared with compounds **2** or **7**, the inhibition found with galantamine (1–10 μM) was significantly higher. IC_50_ values for **2** and **7** were estimated from the non-linear fitted dose-response curves using a Levenberg-Marquardt iteration algorithm. The obtained IC_50_ values were 7.65 μM for **2** and >10 μM for **7**.

#### 2.2.2. Acetylcholine Release

The ability of compounds **2** and **7** to increase K+-evoked [^3^H]ACh efflux in rat hippocampal slices was tested at 1 µM. Compound **2** produced a significant (Student *t*-test *p* < 0.05, *n* = 8–10 per group) increase in [^3^H]ACh efflux (Figure 3). Similarly, **7** (1 µM) was also able to increase ACh release in vitro (Student *t*-test *p* < 0.05, *n* = 8–10 per group) (Figure 3). It is accepted that H_3_R blockade is associated with an increased ACh release, therefore suggesting that these compounds may have therapeutic potential for the symptomatic treatment of cognitive deficits associated with cholinergic deficits [28].

#### 2.2.3. In Vitro Blood-Brain Barrier Permeation Assay (PAMPA-BBB Assay)

Brain permeability is a pivotal feature for central nervous system (CNS)-acting drugs. Thereby, the most potent in vitro AChEI (**2**, **6,** and **7**) were tested using the PAMPA-BBB assay, which measures the passive diffusion of a compound across a porcine brain lipid (PBL)-coated membrane. Compounds with permeation (Pe) values greater than 4 × 10^−6^ cm/s should cross the BBB with no difficulty and reach the CNS. On the contrary, compounds with Pe values below 2 × 10^−6^ cm/s should not cross the BBB. Compounds **6** and **7** showed Pe values (0.32 × 10^−6^ cm/s and 0.47 × 10^−6^ cm/s, respectively, Table 2) below this latter threshold and theoretically should not reach the CNS. However, compound **2** presented a higher Pe value than compounds **6** and **7**, showing a Pe of 2.23 × 10^−6^ cm/s after 20 h. Considering both Pe values, compound **2** was selected to be further characterized. In the same conditions, galantamine exhibited higher values than compound **2** [22].

#### 2.2.4. Novel Object Recognition Test (NORT)

Considering data from H_3_ binding [27], AChE inhibition, and ACh release, compound **2** was selected for the behavioral studies. The AChEI galantamine was used as a reference compound. Cognitive impairments were produced by the administration of the cholinergic antagonist scopolamine. Doses of scopolamine and galantamine were chosen as widely used in the literature. For comparative purposes, compound **2** was tested using the same dose as galantamine. Two-way ANOVA indicated a significant interaction [F(1.40) = 4.947, *p* < 0.05; *n* = 8–10 per group] between scopolamine administration and galantamine/compound **2** treatment on the measurement of discrimination between new and familiar objects in the NORT (Figure 4). Further analysis showed that scopolamine-treated mice exhibited memory impairments, as shown by a significantly lower discrimination index compared with controls that were reversed by either galantamine or compound **2** treatments.

The effects observed in NORT do not appear to be associated with differences in locomotor activity, as total distance traveled in the open field (Table 3) was not affected [F(1.46) = 0.007, *p* = 0.993, *n* = 8–10 per group) by either scopolamine administration or galantamine/compound **2** treatment. In addition, as shown in Table 3, there was no difference in the discrimination index in the sample trial in the NORT associated with scopolamine administration or galantamine/compound **2** treatment [(F(1.46) = 0.007, *p* = 0.993, *n* = 8–10 per group].

#### 2.2.5. ADME Properties Prediction

Many drugs fail due to their pharmacokinetic profiles. The risk of such failures can be minimized by subjecting the compounds to ADME property filters to determine the success of a compound for human therapeutic use. The drug-likeness assessment of the selected compounds was performed by predicting the Lipinski rule of Five, which includes molecular weights (MW < 500), lipophilicity (log Po/w < 5), the number of hydrogen bond acceptors (HBA ≤ 10), and number of hydrogen bond donors (HBD ≤ 5). To investigate the druggability of the most interesting compounds, we performed an” in silico” prediction of the physicochemical properties and human intestinal absorption of compounds **2** and **7** using the pkCSM (https://biosig.lab.uq.edu.au/pkcsm/prediction (accessed on 1 September 2023)) platform [29]. Compound **2** did not violate Lipinski’s rule of five (Table 4), for this derivative was expected to be permeable across the cell membrane, easily absorbed, transported, and diffused. In addition, the prediction of human gastrointestinal absorption was also excellent (Table 3). However, the molecular descriptor values for derivative **7** showed that one of the Lipinski rules was violated, resulting in worse gastrointestinal absorption as well as water solubility.

## 3. Discussion

The term “neurodegenerative diseases” refers to a collection of diverse conditions marked by a progressive, frequently late-onset loss of cognitive abilities brought on by the death of certain cell types that are part of the central nervous system (CNS). These age-dependent disorders have become more common over the past few decades, probably as a result of an increase in the senior population in recent years [30]. Since the most effective treatments can only temporarily slow down signs and symptoms, the rise in overall cases presents several challenges to healthcare systems worldwide [31].

Alzheimer’s disease (AD), which is regarded as a chronic neurodegenerative disease, is the most common kind of dementia. It is a rapidly spreading worldwide epidemic with a clinical description of an irreversible, progressive brain disorder [32]. Of the 47 million people worldwide estimated to have dementia, AD accounts for 60–80% of cases, and it is predicted to double every 20 years to reach 131 million by 2050. Moreover, it is projected that the worldwide expenses associated with dementia care will reach 2 billion dollars by 2030 [33].

Due to the intricate pathophysiological characteristics that underlie AD, the disease is still incurable. Prior studies have concentrated on treatment approaches that combat the supposed causes of neurodegeneration, such as neurofibrillary tangles and the amyloid cascade [34]. Despite many significant findings and promising developments, given the multiple failures of late-stage clinical trials, there are doubts regarding the efficacy of addressing amyloid pathology alone in modifying disease progression [35].

Currently, AD patients receive palliative therapies that enhance quality of life and alleviate symptoms. These therapies concentrate on maintaining acetylcholine-containing forebrain neurons [36] through the use of cholinesterase inhibitors [37], which ultimately raise ACh levels. Only five pharmacological treatment options, currently approved, have been shown in clinical studies to improve the cognitive function of patients with AD. These include one N-methyl-d-aspartate receptor antagonist (NMDA), memantine, and three AChEIs: galantamine, donepezil, and rivastigmine [38]. A fixed-dose combination of memantine and donepezil was eventually approved in 2014 as a fifth treatment option for patients with moderate-to-severe AD who were receiving stable donepezil therapy.

Despite being the most effective target for anti-AD medication development, AChEI has only been used to treat AD symptoms. The finding that the inhibition of histamine receptor H3 (H_3_R) causes precognitive effects has made this receptor another promising target against AD. Therefore, we have generated new dual inhibitors targeting both AChE and H_3_R to study their possible utility in AD.

Specifically, in this study, we investigated a library of sulfonamides and sulfonylureas that were previously designed and synthesized as potential multifunctional anti-AD drugs based on a multi-target-directed ligand strategy.

The two best synthesized candidates (compounds **2** and **7**) were screened for their inhibition potential against AChE and their capacity to induce ACh release. According to the screening test, which was carried out at an inhibition concentration of 0.01–10 µM, the two compounds inhibit the enzyme in different percentages and induce ACh release at a concentration of 1 µM. The results were contrasted with a known anticholinesterase drug (galantamine) as a positive control. Compound **2** exhibited the highest AChEI activity for all concentrations tested. It is remarkable that ACh release was obtained at a concentration that is lower than the IC_50_ of this compound, which suggests that cholinoceptor activation could be implicated in this effect.

Currently, the development of novel anti-AD drugs has a major and serious challenge, such as the permeability of the blood-brain barrier (BBB). As listed in Table 1, compound **2** presented moderate BBB permeation in vitro, suggesting that this derivative could cross BBB through. One of the widely adopted criteria for assessing the drug-like properties of a molecule is Lipinski’s rule of five [29], which relates to lipophilicity, molecular weight, and the number of H-bond donors and acceptors. Noteworthy pharmacokinetic studies showed that compound **2** conformed to Lipinski’s rule of Five, suggesting favorable ADME properties for the development of anti-AD drugs.

Based on the very promising in silico results obtained for compound 2 regarding AChE inhibition capacity and BBB permeability, we conducted an in vivo cognitive test to demonstrate the potential utility of this compound for cognitive improvement. The Novel Object Recognition (NOR) test is based on the innate tendency of rodents to investigate novel objects more thoroughly than they do familiar ones. The decision to look into the novel object shows how learning and recognition memory—more especially, episodic short-term memory—are used. Very remarkably, in our hands, derivative 2 was able to improve cognitive performance in a scopolamine-induced AD mouse model. Of note, the cognitive improvement with compound 2 was as strong as the one observed with galantamine, indicating the potential utility of this new derivative for the treatment of AD symptoms.

In this study, we demonstrated that the combination of histamine H_3_R antagonism and AChE inhibition in a single molecular entity could improve cognitive function and cholinergic neurotransmission, both of which are decreased in AD. Compound **2**, a fragment-like molecule with moderate H_3_R antagonist affinity and a significant increase in acetylcholine release, has exhibited excellent in vivo behavior that is similar to galantamine, suggesting that the proposed dual hypothesis may represent a new target for the treatment of AD.

## 4. Materials and Methods

### 4.1. Chemistry

The compounds were synthesized as previously reported [27]. Briefly, chlorosulfonic acid was added to a solution of (3-bromopropoxy)bencene (*i*) to obtain the intermediate 4-(3-bromopropoxy)benzenesulfonamide (*ii*) by reaction with ammonia. The secondary amines were introduced by nucleophilic substitution in the bromopropooxy fragment that yielded the sulfonamides (**1**–**3**). Finally, the sulfonylureas (**4**–**8**) were obtained by the reaction of sulfonamides with the appropriate arylisocyanates.

### 4.2. Biological Evaluation

#### 4.2.1. Acetylcholinesterase Activity

Male Wistar rats weighing 230–250 g were used. AChE activity was assessed using a colorimetric reported method with minor modifications. Hippocampal tissue was homogenized in 39 volumes of 75 mM sodium phosphate buffer (pH 7.4). A mixture of 2 mL containing the compounds assessed, acetylthiocholine iodide, and 50 μL tissue homogenate was incubated for 8 min. The reaction was then terminated by adding 0.5 mL 3% (*w*/*v*) sodium dodecyl sulphate, followed by 0.5 mL 0.2% (*w*/*v*) 5,5′-dithio-bis(2-nitrobenxoic) acid to produce the yellow anion of 5-thio-2-nitrobenzoic acid. The extent of color production was measured spectrophometrically at 420 nm using Ultospec 3000 (Pharmacia Biotech, Piscataway, NJ, USA). All samples were assayed in triplicate. Results were expressed as percentages of control values.

#### 4.2.2. In Vitro ACh Release

Male Wistar rats weighing 230–250 g were used. K^+^-evoked [^3^H]ACh efflux was measured as previously described [39]. Briefly, the hippocampus was removed and dissected, cut sagittally into 500 µm slices using a McIlwain tissue chopper (The Mickle Laboratory Engineering Co. Ltd., Dorking, UK). Briefly, after labeling the tissue with [^3^H]choline (3 μL/mL, 81 Ci/mmol), aliquots of 4–5 slices were added to each chamber of a Brandel Superfusion-1000 apparatus and superfused with Krebs-Ringer bicarbonate buffer containing the choline reuptake inhibitor hemicholinium-3 (1 μM). Fractions were collected at 3-min intervals for a total of 60 min. At 12 min (S1) and 45 min (S2) after equilibration, the slices were depolarized by changing the superfusion fluid during 6 min to a solution containing 20 mM KCl. Drugs were added 15 min before S2. Tritium content was assayed by liquid scintillation spectroscopy. S1 and S2 were calculated as K+-stimulated tritium increases over basal efflux. Results were expressed as the S2/S1 ratio.

#### 4.2.3. In Vitro Blood-Brain Barrier Permeation Assay

To determine the blood-brain barrier permeability potential of the selected compounds (**2**, **6,** and **7**), the parallel artificial membrane permeability assay (PAMPA) was conducted in Franz diffusion cells (Microette 8910130, Hanson Research, Adelaide, Australia) [40]. Compounds were dissolved in DMSO at 5 mg/mL and diluted to 500 µg/mL with phosphate-buffered saline (PBS)/EtOH (7:3) to make the different stock solutions. Then, a PVDF membrane (pore size 0.45 mm) was coated with 10 µL of porcine brain lipid (PBL) diluted in dodecane solution (20 mg/mL) and placed between the donor and acceptor compartments. 4.5 mL of PBS/EtOH (7:3) was added to the acceptor compartment, and 700 µL of each stock solution was added to the donor cell. After maintaining this structure for 20 h at 25 °C, the concentrations of the tested compounds in the acceptor and donor compartments were measured spectrophotometrically (λ = 239 nm for compound **2**, λ = 261 nm for compound 6, and λ = 236 nm for compound **7**, 8453 UV-Visible Agilent Technologies, Santa Clara, CA, USA). The concentration of the compounds was calculated from the standard curve and expressed as permeability (Pe) by the following formula [41].
Permeability (cm/s): Pe = {−ln [1 − CA(t)/Ceq]}/[A ∗ (1/VD + 1/VA) ∗ t]

A = filter area (0.636 cm^2^), VD = donor cell volume (0.7 mL), VA = acceptor cell volume (4.5 mL), t = incubation time, CA (t) = compound concentration in acceptor cell at time t, CD (t) = compound concentration in donor cell at time t, and Ceq = [CD(t) ∗ VD + VA(t) ∗ VA]/(VD + VA).

#### 4.2.4. Cognitive Evaluation: Novel Object Recognition Test

Male C57BL/6 mice (Harlan, Spain, 8–10 weeks of age) were used. Food and water were available ad libitum for the duration of the experimental procedures, unless otherwise specified. Animals were maintained in a temperature-controlled (21 ± 1 °C) and humidity-controlled (55 ± 2%) room on a 12-h light-dark cycle (lights on at 8:00 h) with free access to food and water. All the experiments were performed in strict compliance with the recommendations of the EU (DOCE L 358/1/18/2/1986) for the care and use of laboratory animals. Adequate measures were taken to minimize the number of animals used in this study as well as their possible suffering.

Drugs were administered (acutely) intraperitoneally (i.p.), 30 min (scopolamine), or 45 min. (galantamine and **2**) before the NORT. Compound **2** (2.5 mg kg^−1^, i.p.), scopolamine (Sigma, St Louis, MO, USA, 1 mg kg^−1^ i.p. and galantamine (Tocris, 2.5 mg kg^−1^ i.p.) were dissolved in a vehicle (0.9% physiological saline) containing Tween-20. Compound **2** and galantamine were injected 45 min before the training session (as described below). Scopolamine was administered 30 min before the training session.

As previously described [42], the open field consisted of a square open field (65 × 65 cm^2^, 45 cm height) made of black wood. On the day before the experiment, the animals were familiarized with the square for 30 min. During this habituation session, horizontal locomotor activity was measured to preclude any motor alteration that could influence cognitive testing. Locomotor activity was measured using a video tracking system (Ethovision 3.0, Noldus Information Technology B.V., Wageningen, The Netherlands) in a softly illuminated room. The tracking system was set to determine the position of the animal five times per second. The total path length (cm) was analyzed. During the first trial (sample phase or training session), two identical objects were placed within the chamber, and the rat was allowed to freely explore for 5 min. Exploration was considered when the head of the rat was oriented toward the object with its nose within 2 cm of the object. One hour later, a second trial took place, in which one object was replaced by a different one, and exploration was scored for 5 min. Results were expressed as the percentage of time spent with the novel object with respect to the total exploration time (discrimination index).

#### 4.2.5. ADME Prediction Properties

Lipinski and GI parameters were calculated using pkCSM software (https://biosig.lab.uq.edu.au/pkcsm/prediction (accessed on 1 September 2023)).

#### 4.2.6. Data Analysis

Different concentrations of the compounds were tested for their ability to inhibit AChE activity. The percentage of AChE inhibition values at these concentrations for compounds **2**, **7**, and galantamine were plotted on Figure 1 using Graph Pad Prism v3.0. The IC_50_ values for **2** and **7** were estimated from the non-linear fitted dose-response curves through a Levenberg-Marquardt iteration algorithm using Origin 2019.

Data from ACh release and behavioral data were analyzed by SPSS for Windows, release 15.0. Normality was checked by the Shapiro-Wilk test (*p* > 0.05). Data from releasing experiments was analyzed by a Student *t*-test (control vs. compound). Behavioural data were analyzed by a two-way analysis of variance ANOVA (scopolamine × treatment), followed by a Student’s *t*-test adjusted by Bonferroni correction.

## Data Availability

Data contained within the article.

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
