# Peer review of "Novel Pitolisant-Derived Sulfonyl Compounds for Alzheimer Disease"

_ijms, 2024, doi:10.3390/ijms25020799_

Round 1

Reviewer 1 Report

Comments and Suggestions for Authors

Authors present the synthesis and characterization of pitolisant-derived sulfonyl compounds for potential Alzheimer disease treatment. The work has scientific soundness and will be of interest to those working in AD drug design. However, there are some points to deserve attention.

Why compounds 6 and 8 were not included in characterization? A difference of 7% in inhibition at 1 uM is small. These compounds should be included at least until PAMPA-BBB assays, this is important because it could be possible that one of these or maybe both showed a positive result in ACh release and BBB permeation. Furthermore, these results contribute to perform a SAR study that increase the discussion and contribution of the work to the knowledge in the search of new drugs against AD under this novel dual hypothesis.

Minor details

What means numbers in parenthesis in figure 1?

Why higher concentrations of compounds were not used in IC50 determination? According to the image a complete sigmoidal curve was not obtained, affecting IC50 value.

Reviewer 2 Report

Comments and Suggestions for Authors

Alzheimer's disease is a neurodegenerative disorder with poor therapeutic treatment. Most of the therapies target cholinergic neurotransmission in the CNS. The Authors selected Compound 2 with cholinesterase inhibitory and H3 receptor antagonist/partial agonist characteristics with acceptable pharmacokinetics parameters as a potential compound for drug selecton. Compound 7 showed low BBB penetration. Comments: Add more information about the effects of compounds on H3R binding. Also muscarinic and nicotinic receptor binding assay would support the view of the Authors. Add IC50 value of gallantamine in AChE assay, AChE inhibitory effects of compounds 2 an 7needs to be measured in higher than 10 uM also. Compounds 2 and 7 increased [3H]ACh release in a concentration in which effect in AChE inhibition was marginal: effects on cholinoceptors may have importance in this effect. Express BBB data as mean+/-S.E.M. Add own or literature data for galantamine  BBB penetration for comparison. Table 2: add n value, mean+/-S.D., S.E.M.? Addition of a H3 receptor antagonist/partial agonist reference compound into the tests used would strengthen the Authors' conclusion.

Comments on the Quality of English Language

Line 117: an essential

Reviewer 3 Report

Comments and Suggestions for Authors

In the manuscript 'Novel Pitolisant-Derived Sulfonyl Compounds for Alzheimer's Disease,' the authors aimed to describe the effectiveness of the sulfonyl derivative of pitolisant as an HT3 receptor antagonist. As per my work applications, I refrain from commenting on the design and chemical syntheses carried out. However, I can provide my opinion on other aspects of the manuscript. In my experience, the manuscript appears to be well-organized and complete with methodology. However, I believe that it would be beneficial to include more information regarding the use of pitolisant (treatment of narcolepsy), perhaps in the introduction and discussion, even if the authors wanted to focus only on the antiacetylcholinesterase effect. Additionally, I think that the references of the entire manuscript should be increased by adding some information regarding the pathology of Alzheimer's, sex, and gender. These aspects cannot be overlooked, especially from the perspective of human treatment.

I believe there isn't anything else to add."

Round 2

Reviewer 1 Report

Comments and Suggestions for Authors

The manuscript was corrected according to the suggestions. Therefore, I recommend its publication.